# Bioinformatics Analysis Highlight Differentially Expressed CCNB1 and PLK1 Genes as Potential Anti-Breast Cancer Drug Targets and Prognostic Markers

**DOI:** 10.3390/genes13040654

**Published:** 2022-04-07

**Authors:** Leiming Fang, Qi Liu, Hongtu Cui, Yunji Zheng, Chengjun Wu

**Affiliations:** 1School of Biomedical Engineering, Dalian University of Technology, Dalian 116024, China; flming1018@foxmail.com (L.F.); liuqi_dut@163.com (Q.L.); cuihongtu@mail.dlut.edu.cn (H.C.); 2School of Pharmacy, Binzhou Medical University, Yantai 264003, China; yunji.zheng@med.lu.se

**Keywords:** breast cancer, CCNB1, DEGs, hub genes, PLK1

## Abstract

Breast cancer is one of the most common malignant tumors in women worldwide. Early diagnosis, treatment, and prognosis of breast cancer are global challenges. Identification of valid predictive diagnosis and prognosis biomarkers and drug targets are crucial for breast cancer prevention. This study characterizes differentially expressed genes (DEGs) based on the TCGA database by using DESeq2, edgeR, and limma. A total of 2032 DEGs, including 1026 up-regulated genes and 1006 down-regulated genes were screened. Followed with WGCNA, PPI analysis, GEPIA 2, and HPA database verification, thirteen hub genes including *CDK1*, *BUB1*, *BUB1B*, *CDC20*, *CCNB2*, *CCNB1*, *KIF2C*, *NDC80*, *CDCA8*, *CENPF*, *BIRC5*, *AURKB*, *PLK1*, *MAD2L1*, and *CENPE* were obtained, and they may serve as potential therapeutic targets of breast cancer. Especially, overexpression of *CCNB1* and *PLK1* are strongly associated with the low survival rate of breast cancer patients, demonstrating their potentiality as prognostic markers. Moreover, *CCNB1* and *PLK1* are highly expressed in all breast cancer stages, suggesting that they could be further studied as potential drug targets. Taken together, our study highlights *CCNB1* and *PLK1* as potential anti-breast cancer drug targets and prognostic markers.

## 1. Introduction

Breast cancer is a malignant tumor arising from the deterioration of lesions in the epithelial tissue of the breast. Approximately 1.4 million people worldwide are diagnosed with breast cancer, causing about 500,000 deaths each year [1]. As the biggest threat to women’s life, breast cancer has the highest incidence and second highest mortality rate among the female population [2,3]. At present, the number of breast cancer patients is still growing at an extremely fast speed. About 2.3 million new breast cancer cases were diagnosed worldwide in 2020. Breast cancer surpassed lung cancer as the most diagnosed cancer worldwide for the first time and accounted for 11% of new cancer cases worldwide in 2020 [4]. Therefore, it is essential to reveal the pathogenesis of breast cancer and discover potent new biomarkers for breast cancer diagnosis, treatment, and prognosis.

In recent years, with an in-depth study of the molecular mechanism of breast cancer, researchers have reported a series of abnormally expressed genes involved in the occurrence and development of breast cancer. For example, the expression of *RBBP7* and *BIRC5* in ER-positive ductal carcinoma in situ (DCIS) is significantly higher than in invasive ductal carcinoma (IDC) [5]. Overexpression of the *BCL11A* gene is associated with triple-negative breast cancer (TNBC) [6]. *TMEM45A*, *FAT1*, and *DST* play significant roles in transforming carcinoma in situ into invasive carcinoma [7]. Overexpression of *FOXK1* promotes proliferation, migration, and invasion of breast cancer cells [8]. Regulation of *GATA-3* and *FOXA1* genes expression in HR-positive/HER2-negative breast cancer contributes to cancer treatment [9]. In addition, isomers have been reported to play key roles in breast cancer processing. For example, *BRD4-S* is a short isomer with less content in *BRD4* functioning as a carcinogen protein, while the isomer *BRD4-L* can inhibit breast cancer cells’ formation, proliferation, and migration [10]. *AKT* has two antagonized isomers. *AKT1* promotes cell proliferation and reduces the migration of breast cancer cells, while *AKT2* promotes migration and invasion of cancer cells [11]. The discovery of these abnormally expressed genes paves a novel way for drug targets development. However, the mechanism of these abnormally expressed genes remains to be studied. Therefore, it is urgent to continuously search for abnormally expressed genes in breast cancer that can be used as diagnostic markers and drug targets.

At present, bioinformatics technology is widely used to predict the functions of key genes involved in breast cancer progression. Jin et al. found that *FOXC1* plays a vital role in human basal-like breast cancer through the analysis of public databases [12]. Cheng et al. found that increased *TIMP-1* expression level is strongly associated with a poor prognosis of TNBC [13]. Therefore, we wish to screen abnormally expressed genes that can be used as drug targets and prognostic markers from public databases using bioinformatics analysis.

## 2. Materials and Methods

### 2.1. Data Acquisition and Preprocessing

This study is based on the following technical route (Appendix A). The acquisition of breast cancer sample data (HTSeq-Counts) was performed in the GDC (Genomic Data Commons) data portal of the TCGA (https://cancergenome.nih.gov/, accessed on 6 March 2022) database. We finally obtained the gene expression data of 1215 tissue samples, of which 103 were normal, and 1102 were cancer tissue samples. The following three methods were used to screen the low expression genes in the bioinformatics analysis software RStudio.

Building a DGElist object, and then filtering out the low expression genes using the FilterbyExpr function attached to the edgeR toolkit;For the convenience of analysis, the expression levels of different genes in the database have been assigned. The genes’ average expression levels bigger than 1 in each sample were screened;The expressed genes were screened in 75% of the samples.

### 2.2. Identification of Differentially Expressed Genes

This study used DESeq2, edgeR, and limma to screen differentially expressed genes (DEGs) between normal breast tissue and tumor tissue samples. The corresponding normalization processing is carried out according to the median of ratio of DESeq2 and edgeR’s TMM. The screening conditions were |log2 (FC)| > 1.5 and *p* ≤ 0.05.

### 2.3. Weighted Genes Correlation Network Analysis

The obtained DEGs were analyzed by weighted genes correlation network analysis (WGCNA) [14]. The steps of WGCNA mainly include gene co-expression similarity matrix calculation, adjacency function calculation, soft threshold selection, topological overlap matrix, heterogeneity matrix calculation, dynamic branch cutting calculation of gene module, and correlation analysis between gene modules and sample clinical information.

### 2.4. Construction of Protein-Protein Interaction Network

The protein-protein interaction (PPI) network was constructed using the STRING database. The STRING database excluded proteins with weak connections with other proteins in the network by setting the minimum interaction score. The minimum interaction score was set to “the highest confidence (0.900)” to obtain the module genes’ protein interaction network diagram.

### 2.5. Screening of Core Modules and Hub Genes

The constructed PPI network was imported into Cytoscape analysis software, and the core modules and hub genes were screened by MCODE and CytoHubba plug-ins. The setting parameters of MCODE plug-in were: degree cut-off = 2, node density cut-off = 0.1, node score cut-off = 0.2, k-core = 2, and maximum depth = 100. This study used three algorithms in the CytoHubba plug-in: MCC (Maximal Clique Centrality), Degree, and Closeness to obtain the hub gene. Each algorithm selected twenty genes with the highest score and the intersection of three results was adopted to obtain the hub genes.

### 2.6. Expression Analysis

The hub gene expression in normal and cancer tissues was obtained from the GEPIA 2 database (http://gepia2.cancer-pku.cn/index.html, accessed on 6 March 2022). The difference of hub gene expression was observed to inspect the screening results from the TCGA database. The protein immunohistochemical staining results obtained from the HPA database (https://www.proteinatlas.org/, accessed on 6 March 2022) were used to verify the expression levels of the hub gene in normal and cancer tissues. The protein immunohistochemical staining levels were divided into “not detected”, “low”, “medium”, and “high”.

### 2.7. Survival Analysis

ROC (receiver operating characteristic) analysis and K-M uni-variate analysis were performed to verify the possibility of hub genes as tumor diagnostic markers and therapeutic targets.

### 2.8. Kaplan-Meier Plotter

“Auto select best cutoff” was used to identify the stratification threshold and evaluate the overall survival (OS), post-progression survival (PPS), and recurrence-free survival (RFS) of breast cancer patients. The corresponding 95% confidence interval and *p*-value were set.

### 2.9. Analysis of Candidate Targets

GEPIA 2 database was used to investigate the relationship between target genes expression and stage grades, the expression map of target genes in other cancers, and the isoforms expression distribution of target genes.

### 2.10. Gene Enrichment Analysis

The enrichment analysis results were verified from three aspects:Gene Ontology (GO) and KEGG enriched module genes obtained by WGCNA;DAVID database was used to conduct GO and KEGG enrichment analysis on the modules with the highest scores screened by the plug-in MCODE;GEPIA 2 database was used to screen ten genes that similar to each hub gene’s expression pattern. The Metascape (https://metascape.org/, accessed on 6 March 2022) was chosen for enrichment analysis.

### 2.11. Tools

All bioinformatics analysis in this study used R (https://www.r-project.org/, accessed on 6 March 2022) language and multiple data packets, including “tcgabiolinks”, “ggplot2”, “Rio”, “rjson”, “edger”, “deseq2”, “limma”, “pheatmap”, “factoextra”, “factominer”, “venndiagram”, “WGCNA”, “survival”, “surfminer”, “proc”, “clusterProfiler”, “enrichplot”, “topGO”, “ReactomePA”, “pathview”, “AnnotationHub”, and “AnnotationDbi”.

## 3. Results

### 3.1. Identification of 1026 Up-Regulated Genes and 1006 Down-Regulated Genes

To reduce the impact of the batch effect on sample screening and subsequent analysis, we excluded five duplicate samples to obtain a multidimensional gene expression matrix including 20,546 genes from 113 normal breast tissue and 1097 cancer tissues’ gene expression profiles. Principal component analysis (PCA) was adopted to reduce the high-dimensional gene expression data into several principal components. By analyzing the similarity between principal components, the overall similarity between samples was obtained. The similarity of samples lower than 30% indicated that the gene expression level between normal and tumor samples was significant. If the sample similarity was higher than 30%, the batch correction was required. The PCA results indicated that the general gene expression levels of the normal and tumor groups were significantly different. The first principal components explained 19.9% of the total variance, while the second principal component was 9% (Figure 1A). Since the significant difference in the gene expression between normal and tumor samples was observed, we wished to identify DEGs using DESeq2, edgeR, and limma. DESeq2, edgeR, and limma are the gold standards for the transcriptome differential expression analysis [15,16,17]. Considering the analysis error caused by using any of the three methods alone, the intersection of the three methods was used to reduce the errors [18]. DESeq2 identified 1625 up-regulated and 1121 down-regulated genes; edgeR identified 1667 up-regulated genes and 1206 down-regulated genes; and limma identified 1142 up-regulated genes and 1554 down-regulated genes (Figure 1B–D). Integration of the results generated from these three methods identified 1026 up-regulated genes and 1006 down-regulated genes (Figure 1E–F).

### 3.2. Identification of 693 Module Genes Correlated with Breast Cancer

Earlier studies indicated that clinical phenotypes impacted breast cancer treatment. To further study the correlation of clinical phenotype with identified DEGs, we firstly constructed a corresponding weighted gene co-expression network. A total of 1209 samples were clustered as a clustering tree, as shown in Appendix A. Afterward, we applied weighted gene co-expression network analysis (WGCNA) to reveal the correlation of DEGs with clinical phenotypes, including gender (male, female), survival time (0−8605 days), age (26−90 years), cancer stage (I, II, III, IV, X), TNM stage (M0, M1, MX; N0, N1, N2, N3, NX; T1, T2, T3, T4, TX), survival status (death, survival), and breast cancer infection status (positive, negative). 

WGCNA was further applied for cluster analysis to highlight module genes. To make the network constructed by WGCNA approximate to a scale-free network, we used the proximity function of WGCNA in R to obtain the soft threshold β= 8 (scale-free R^2^ = 0.9) (Appendix A). All selected genes were clustered by using the dynamic tree-cutting algorithm based on the topological overlap matrix (TOM), the minMoudleSize (minimum number of genes in the module) was set to 30, and the mergeCutHeight (minimum distance of merging module genes) was used to merge modules that distance was less than the set value (0.25). The clustering tree was divided into six modules (Appendix A). Genes that were not included and correlated with any module were classified as a gray module, removed from subsequent analysis. The number of genes in each module were shown in Appendix A. We analyzed the interaction of the five modules and generated the network heat map, which indicated the high degree of independence between modules and the relative independence of gene expression in each module (Figure 2A).

Following the construction of the WGCNA network, we wished to investigate the correlation of DEGs with clinical phenotypes. We calculated the MS (module significance), which indicated the correlation of each module trait and association with the different phenotypic traits of breast cancer. We observed that the module characteristic genes (ME) of turquoise (r = 0.72, *p* = 1 × 10^−196^) and brown (r = 0.72, *p* = 1 × 10^−194^) modules had a stronger correlation with breast cancer infection status than other modules, which contained a total of 693 genes (Figure 2B). 

### 3.3. A Total of Fifteen Hub Genes Were Selected by the PPI Network

The interaction between proteins profoundly affected all aspects of activities in cells. To explore the core modules and hub genes that played the greatest role in modular genes, the STRING database was applied to construct a PPI network of these 693 genes (Figure 3A). Cytoscape was utilized to visualize the PPI network (Figure 3B). The plug-in MCODE in Cytoscape was used to identify the dense connection region of the molecular interaction network based on each molecular’s connection data, of which had a high possibility of participation in biological regulation. Finally, sixteen important modules were obtained (Appendix A), and the module with the highest score was visualized (Figure 3C).

Since too many hub genes were included in the modules, it was difficult to understand the role of these hub genes in breast cancer progression. Therefore, we wished to highlight key hub genes from complex interactive networks by using plug-in CytoHubba in Cytoscape, which can use different algorithms to calculate the score of each node in the network, and then sort it. Evaluation of key hub genes were decided by MCC, Degree, and Closeness. We selected the candidates of key hub genes, as shown in Appendix A. Moreover, we constructed the network map of these highlighted key hub genes, as shown in Figure 3D–F. Finally, fifteen hub genes were obtained, including *CDK1*, *BUB1*, *BUB1B*, *CDC20*, *CCNB2*, *CCNB1*, *KIF2C*, *NDC80*, *CDCA8*, *CENPF*, *BIRC5*, *AURKB*, *PLK1*, *MAD2L1*, and *CENPE*.

### 3.4. Fifteen Hub Genes Were Highlighted as Potential Therapeutic Drug Targets

The differential expression of the selected fifteen hub genes in normal tissues and cancer tissues were detected by GEPIA 2 and HPA database to analyze whether hub genes could be used as breast cancer diagnostic markers. The results showed that the expression levels of fifteen hub genes in cancer tissues were higher than that in normal tissues, further suggesting that these fifteen hub genes may play key roles in the development of breast cancer (Figure 4). The protein expression level of hub genes was verified using the HPA database. Since lacking data for *BUB1*, *BUB1B*, and *NDC80* genes in the HPA database, we showed only twelve key hub genes’ results (Appendix A). The expression of hub genes were specified as “high”, “medium”, “low”, and “not detected”. Ten hub genes (*CDK1*, *CCNB2*, *CCNB1*, *CDCA8*, *CENPF*, *BIRC5*, *AURKB*, *PLK1*, *MAD2L1*, and *CENPE*) were highly expressed in cancer tissues compared to normal tissues. However, the antibody staining of *CDC20* and *KIF2C* both showed medium in breast cancer and normal tissues, so no significant difference was observed in the expression of *CDC20* and *KIF2C* genes. 

To further verify the predictive performance of the highlighted key hub genes as potential drug targets, we conducted ROC analysis and calculated the area under curve (AUC) to compare the expression level of these fifteen key genes in normal and tumor tissue. The results indicated that the AUC values of fifteen key hub genes were more than 95%, which further suggested the expression of these fifteen key hub genes can be distinguished from tumor tissues to normal tissues (Figure 5). Our observation was consistent with GEPIA 2, and HPA databases’ results indicated that these hub genes might be potential tumor molecular markers of breast cancer.

### 3.5. Highly Expressed CCNB1 and PLK1 in Breast Cancer Were Associated with a Low Survival Rate

To explore the relationship between the expression of fifteen hub genes and the survival rate of breast cancer patients and whether they can be used as potential targets for treatment and prognosis, we conducted a survival analysis on selected hub genes. The results suggested that only *CCNB1* and *PLK1* could be potential therapeutic targets and prognostic markers (Figure 6A,B). However, the other 13 genes were not statistically significant (*p* > 0.05) (Appendix A). We wished to further evaluate the *CCNB1* and *PLK1* as therapeutic targets and prognostic markers. Kaplan–Meier plotter was applied to analyze the correlation of survival rate with *CCNB1* and *PLK1* expression. The results demonstrated that the high expression of the *CCNB1* group’s overall survival rate declined faster than the patients that *CCNB1* was less expressed from 0 to180 months. However, the overall survival rate of both groups were nearly the same at the 180th month. After 180 months, the survival rate of the low-expression group remained at 0.7, while the survival rate of the high expression group decreased remarkably until remained at 0.4 (Figure 6C–E). In addition, we also investigated the post-progression survival rate and recurrence-free survival rate. In line with the observation of overall survival rate, *CCNB1* high-expression group’s post-progression survival rate and recurrence-free survival rate significantly decreased compared to the low-expression group. Similarly, the analysis of all three survival rates of patients with high-expression *PLK1* were lower than the *PLK1* low-expressed group (Figure 6F–H). The significant correlation between the high expression of *CCNB1* and *PLK1* and the low survival rate of patients implied the possibility of *CCNB1* and *PLK1* as prognostic markers and as targets to screen expression inhibitors during treatment.

To further explore the role of *CCNB1* and *PLK1* in the development of breast cancer, we used the GEPIA2 database to analyze the expression levels of *CCNB1* and *PLK1* at different clinical stages. The results indicated that the expression levels of *CCNB1* and *PLK1* in stage III were higher than that in patients in other stages (Appendix A). Previous studies reported that different isoforms played vital roles in breast cancer development. Interestingly, *CCNB1* has seven known isoforms, while *PLK1* has nine isoforms. We selected the GEPIA2 database to analyze the expression of *CCNB1* and *PLK1* isoforms in breast cancer patients. The results indicated three *CCNB1* isoforms, including *CCNB1*-001, *CCNB1*-003, and *CCNB1*-006, and the isoform *PLK1*-001 of *PLK1*, were highly expressed in breast cancer samples (Appendix A).

In addition, we also used GEPIA2 to investigate the expression of *CCNB1* and *PLK1* in a variety of other cancers, and the results demonstrated that they were highly expressed in most tumor tissues. Especially, they were significantly over-expressed in lung cancer, gastric cancer, rectal cancer, and brain cancer tissues. This observation further illustrated that the over-expression of *CCNB1* and *PLK1* were of great significance for the occurrence of other cancers (Appendix A).

### 3.6. CCNB1 and PLK1 Were Enriched in Breast Cancer Cell Division and Cell Cycle-Related Pathways

We conducted enrichment analysis in three aspects to carry out the multi-dimensional cross-analysis. We firstly performed GO and KEGG enrichment analysis on 693 module genes obtained from WGCNA (Figure 7A,B). KEGG analysis showed that the genes mainly enriched in “cell cycle,” “oocyte meiosis,” “human T-cell leukemia virus one infection,” and “progesterone-mediated oocyte maturation.” The GO enrichment analysis results showed the genes in BP (biological process) mainly enriched in “organelle fission,” “nuclear division,” and “chromosome segregation.” CC (cellular component) was mainly enriched in the “chromosomal region”, “spindle”, “chromosome, centromeric region”, and “condensed chromosome” and other regions. MF (molecular function) mainly enriched in “tubulin-binding”, “ATPase activity”, and “microtubule-binding”.

The DAVID database was used to perform GO and KEGG analysis on the genes of the core module with the highest score to investigate the core module’s main action pathway (Appendix A). The analysis results were shown in Appendix A. The five most enriched pathways obtained are “GO:0007062-sister chromatid cohesion”, “GO:0051301-cell division”, “GO:0007067-mitotic nuclear division”, “GO:0000777-condensed chromosome kinetochore”, and “GO:0005829-cytosol”.

Since the number of hub genes is small, we selected ten genes with similar expression patterns for each hub gene in the GEPIA 2 database to reduce the errors in the enrichment analysis. We obtained a total of 150 genes; the enrichment analysis of these genes was performed using Metascape to study the gene action pathways related to the expression pattern of the hub gene and compared with the previous module gene action pathways (Figure 7C). The results showed that similar genes of the hub gene were mainly enriched in “cell division,” “spindle,” “chromosome, centromeric region,” “regulation of cell cycle process,” and “spindle pole.”

## 4. Discussion

Breast cancer is a malignant tumor with high molecular heterogeneity. Since the late 1970s, breast cancer has been the leading cause of women’s death [1]. The formation, development, and deterioration of breast cancer are affected by multiple factors such as genetic factors, therapeutic factors, and environmental factors. For a long time, the clinical treatment of breast cancer has been based on surgery, supplemented by chemotherapy, endocrine therapy, radiotherapy, and traditional Chinese medicine treatment. The discovery of molecular markers and drug targets paved the way for a novel treatment based on molecular targets. At present, gene targeting therapy for breast cancer is mainly focused on the treatment of oncogenes, tumor suppressor genes, and tumor angiogenesis genes of breast cancer. In addition, especially indirect therapy will be carried out for breast cancer immune genes, drug-sensitive genes, and multi-drug resistance genes [19]. The wide use of molecular targets in diagnosis, treatment, and prognosis encourages researchers to find better molecular targets. It is essential to understand the molecular basis playing key roles in the development and progression of breast cancer.

In this study, we obtained genes expression data of 1097 breast cancer samples and 113 normal tissues from the TCGA database. The analysis of DESeq2, edgeR, and limma indicated 1026 up-regulated genes and 1006 down-regulated genes in breast cancer tissue samples. Since the samples contained a large number of DEGs, we only focused on the part with the highest correlation when analyzing the correlation between traits and DEGs. We highlighted two modules, which were the most relevant to breast cancer status, including 693 module genes. Then, the PPI network containing 693 genes was constructed using a STRING database, and sixteen core modules were obtained. The hub genes in the PPI network were selected by calculating three algorithms: MCC, Degree, and Closeness. A total of fifteen hub genes (*CDK1*, *BUB1*, *BUB1B*, *CDC20*, *CCNB2*, *CCNB1*, *KIF2C*, *NDC80*, *CDCA8*, *CENPF*, *BIRC5*, *AURKB*, *PLK1*, *MAD2L1*, and *CENPE*) were obtained.

This was followed by the detection of the fifteen hub genes expression level in normal and tumor tissues through the GEPIA2 database. The results indicated that all hub genes were highly expressed in tumor tissues. HPA database was selected to further verify the expression level of these hub genes in tumor tissues. *BUB1*, *BUB1B*, and *NDC80* were excluded because they were not recorded in the HPA database. The protein immunohistochemical staining indicated the levels of *CDC20* and *KIF2C* were nearly the same in normal and cancer tissues, which was inconsistent with the result that the expression levels of *CDC20* and *KIF2C* in tumor tissues were higher than normal tissues. This may be due to the sample size of *CDC20* and *KIF2C* in the HPA database being too small, post-transcriptional regulation, or any other mechanisms. This may be our future interest remained to be studied. However, the other ten hub genes were highly expressed in cancer tissues, which was consistent with the analysis of the GEPIA2 database. 

ROC analysis of fifteen hub genes showed that the AUC values of fifteen hub genes were all greater than 95%, suggesting that these fifteen hub genes might have a better chance to be tumor diagnostic markers. However, since the samples we obtained were already in the cancer stage, the possibility of them as diagnostic markers of pre-cancer remain to be investigated. Further experimental and clinical verification of their feasibility as diagnostic markers of breast cancer would be needed in the future. In addition, the Kaplan–Meier survival analysis of fifteen hub genes demonstrated that the high expression of *CCNB1* and *PLK1* was significantly correlated with the low survival rate of breast cancer patients. The 5- and 10-year overall survival rates of *CCNB1* and *PLK1* highly expressed patients were about 22% lower than those patients low-expressed. Furthermore, the survival rate of *CCNB1* and *PLK1* highly expressed group was dramatically declined after 15 years, which was about 43% lower than the low-expressed group. We also investigated the relationship between *CCNB1* and *PLK1* and the clinical stage using the GEPIA2 database. The results indicated that *CCNB1* and *PLK1* expression level in stage III were higher than other stages, suggesting they might play special roles in the development of breast cancer, especially in stage III. Previous studies have found that some protein isoforms play an important role in the development of breast cancer. For example, *BRD4* and *AKT* are important for the proliferation of breast cancer cells, and their isomers are antagonized to each other. Interestingly, *CCNB1* and *PLK1* have multiple isoforms. We analyzed the expression levels of their isoforms and found that the expression levels of *CCNB1*-001, *CCNB1*-003, *CCNB1*-006, and *PLK1*-001 in breast cancer tissues were higher than those in normal tissues. In later studies, we will further explore the role of their isoforms in the development of breast cancer. 

As an important member of the cyclin family, *CCNB1* plays an important role in mitosis initiation and regulation. *CCNB1* accumulates in the S phase and reaches the maximum level during mitosis. Afterward, *CCNB1* is degraded rapidly when the cell cycle transits from metaphase to anaphase. At present, the role of *CCNB1* in a variety of cancers has been studied. For example, the silencing of *CCNB1* in pancreatic cancer cells promotes cell senescence, inhibiting cell proliferation and promoting cell apoptosis [20]. The role of *CCNB1* in breast cancer progression is also reported. For instance, Liang Fang et al. reported that circ-*CCNB1* inhibits *p53*, thereby promoting the occurrence and development of breast cancer [21]. The combination of *CCNB1* and paclitaxel increases the apoptosis of breast cancer cells and enhances paclitaxel’s antiproliferative effect [22]. Moreover, *CCNB1* is a highly reproducible prognostic marker in lymph node negative breast cancer [23]. *PLK* family is an essential mitotic regulator, controlling the termination of mitosis by modulating APC production (anaphase-promoting complex) and regulating the coordination of cytokinesis in space and time [24]. *PLK1* is a highly conserved member of the *PLK* family. It plays a vital role as a regulator of cell division during mitosis. *PLK1* is generally enriched in mitotic centrosomes, kinetochores, and cytokinesis midbody, which enable *PLK1* to phosphorylate specific downstream targets, thereby regulating mitosis [25]. At present, researchers elucidate the roles of *PLK1* in the development of breast cancer. Researchers have shown the anti-sense inhibitors, that target *PLK1* inducing a synergistic effect of taxane and paclitaxel, thereby enhancing the sensitivity of breast cancer cells to these drugs in vivo and in vitro [26]. In addition, researchers have reported the transcriptional action of *PLK1* on the regulation of ER mediated by human breast cancer cells, and proposed the mechanism of *PLK1* as a mediator of interphase transcription regulators in animal mammalian tumors [27]. Above all, we concluded that continuous tracking of the expression levels of *CCNB1* and *P**LK1* was of great significance in improving patients’ prognosis and survival rate. Moreover, the identification of inhibitors that targeting *CCNB1* and *PLK1* might provide a new idea for treating breast cancer.

Although the other thirteen hub genes were not significantly correlated with prognosis, their abnormal expression might play an important role in the occurrence and development of breast cancer. For example, blocking *CDK1* expression combined with other therapies has shown strong anti-cancer effects for breast cancer patients [28]. Knockdown of *CDC20* results in inhibition of metastatic MDA-MB-231 migration in breast cancer cell lines [29]. *CCNB2* overexpression stimulates proliferation in vitro and in vivo in three negative breast cancer cells [30]. Although the role of other hub genes in breast cancer has not been specifically reported, studies have shown that they play an essential role in other cancers, such as *BUB1* promotes proliferation by activating *SMAD2* phosphorylation [31]. The injury of BUB1B has a destructive effect on the viability and tumorigenicity of cancer cells [32]. The *TBX15*/miR-152/*KIF2C* pathway regulates adriamycin resistance in breast cancer by promoting *PKM2* ubiquitination [33]. High expression of *NDC80* promotes the malignant progression of colorectal cancer [34]. Knocking down *CDCA8* inhibits the proliferation of bladder cancer cells and enhances apoptosis [35]. *CENPF* regulates cancer metabolism by regulating pyruvate kinase M2 phosphorylation signal transduction [36]. *BIRC5* directly regulates apoptosis and mitosis in cancer cells during tumorigenesis and tumor metastasis [37]. *AURKB* promotes the proliferation of gastric cancer cells in vitro and in vivo [38]. Abnormal expression of *MAD2L1* induces chromosome instability and aneuploidy in cells to promote tumor formation [39]. After knocking down the expression of *CENPE*, the proliferation of lung cancer cells is inhibited [40]. The above evidence suggests that these hub genes play an important role in the occurrence and development of different cancers. If they can be used as clinical targets need to be further studied.

To explore the pathways and regions that the hub genes are involved in, we performed three levels of GO and KEGG enrichment analysis on 693 module genes obtained from WGCNA. The enrichment analysis of these three aspects showed that the above genes were enriched in the “cell cycle” and “cell division” (including nuclear division, organelle division, and chromosome division). The research on the cell cycle and cell division of tumor cells has always been the hot spot. For example, Katrien et al. found that the *P13K* pathway is significantly related to proliferation, cell division, cell differentiation, and apoptosis and may help develop new breast cancer drug therapy combined with trastuzumab [41]. Meanwhile, *CCNB1* and *PLK1* also participate in cell cycle and cell division and function as a regulator of cell division during mitosis, which corroborates results of the previous analysis. In addition, KEGG analysis highlighted a special pathway, which is human T-cell leukemia virus 1 infection. Human T cell leukemia virus 1 (HTLV-1) is a pathogenic retrovirus related to adult T cell leukemia/lymphoma. At present, no research has proved that HTLV-1 is related to the occurrence and development of breast cancer, which may provide new ideas for the pathogenesis of breast cancer.

In summary, we screened DEGs based on the TCGA database and discussed their functions and pathways, which may be related to the occurrence and development of breast cancer. We concluded that fifteen hub genes might be a potential diagnosis target for breast cancer. Especially, we highlighted CCNB1 and PLK1 genes may be potential therapeutic targets and prognosis markers. These preliminary findings will pave the foundation for our future experimental investigation.

## Figures and Tables

**Figure 1 genes-13-00654-f001:**
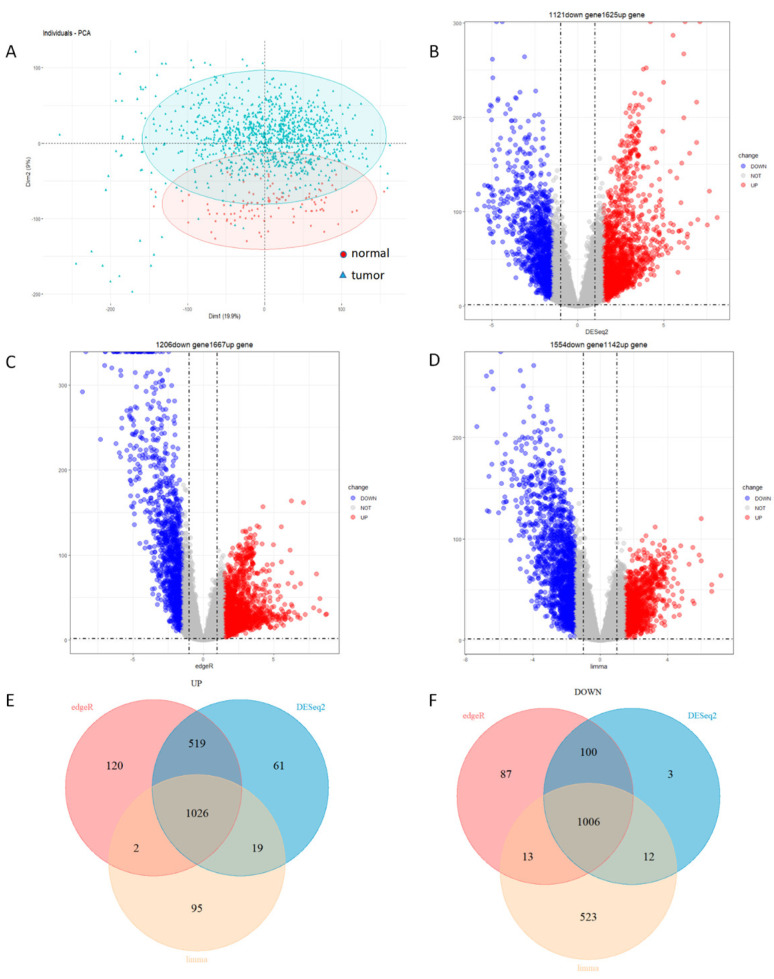
PCA and differential expression analysis. (**A**) PCA analysis of breast cancer samples. The red circle represents the normal tissue, and the blue triangle represents the breast cancer tissue. The x−axis “Dim1” is the similarity between the first principal component of cancer tissue sample and normal tissue sample; the y−axis “Dim2” is the similarity between the second principal component. (**B**) Volcano plots of DEGs screened by DESeq2. (**C**) Volcano plots of DEGs screened by edgeR. (**D**) Volcano plots of DEGs screened by limma. The abscissa in (**B**–**D**) is the logarithm of fold change. The ordinate is −log10 *p* value. The magnitude of the numerical value is positively correlated with significance. Red represents up−regulated genes; blue represents down−regulated genes; and grey represents none significant. (**E**) Venn diagrams of up-regulated genes. (**F**) Venn diagrams of down-regulated genes. (**E**,**F**) demonstrate the number of genes intersected by different methods.

**Figure 2 genes-13-00654-f002:**
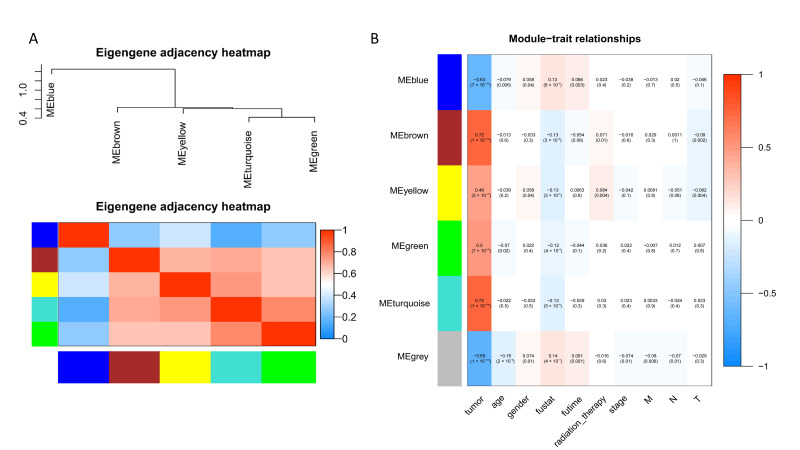
WGCNA analysis results. (**A**) Visualization of module gene network. The upper part is the cluster between modules, and the lower part is the correlation between modules. Color represents the degree of correlation. (**B**) Heatmap of the correlation between ME and phenotype of BRCA. Color represents the correlation between module gene and phenotype, positive value represents positive correlation and negative value represents negative correlation. It also expresses the level of significance. *p* < 0.05 was considered statistically significant. “T”, “N”, and “M” in the x-axis are the TNM stages of breast cancer. “T” is the size and stage of the tumor, “N” is the metastasis of lymph nodes, and “M” is the tumor metastasizes to the distance.

**Figure 3 genes-13-00654-f003:**
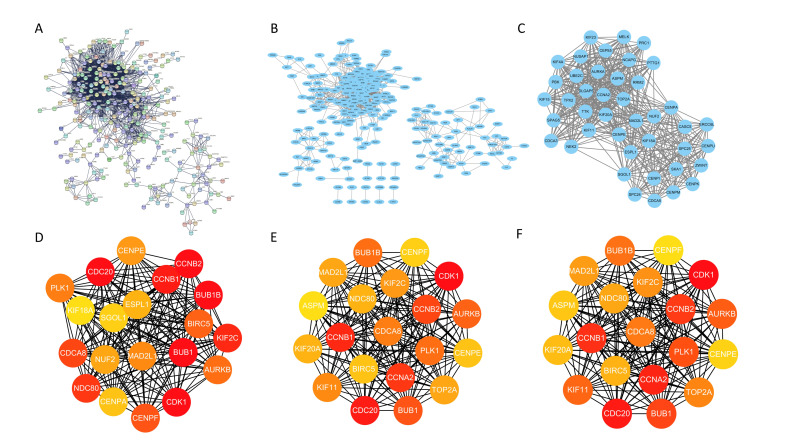
(**A**) PPI network constructed by STRING database. PPI network diagram is composed of nodes and edges. Each node represents a protein, and the connection represents the interaction between proteins. (**B**) Visualization of PPI network in Cytoscape. (**C**) Visualization of the core module with the highest score. (**D**) Hub genes obtained according to MCC. (**E**) Hub gene obtained according to Closeness. (**F**) Hub gene obtained according to Degree.

**Figure 4 genes-13-00654-f004:**
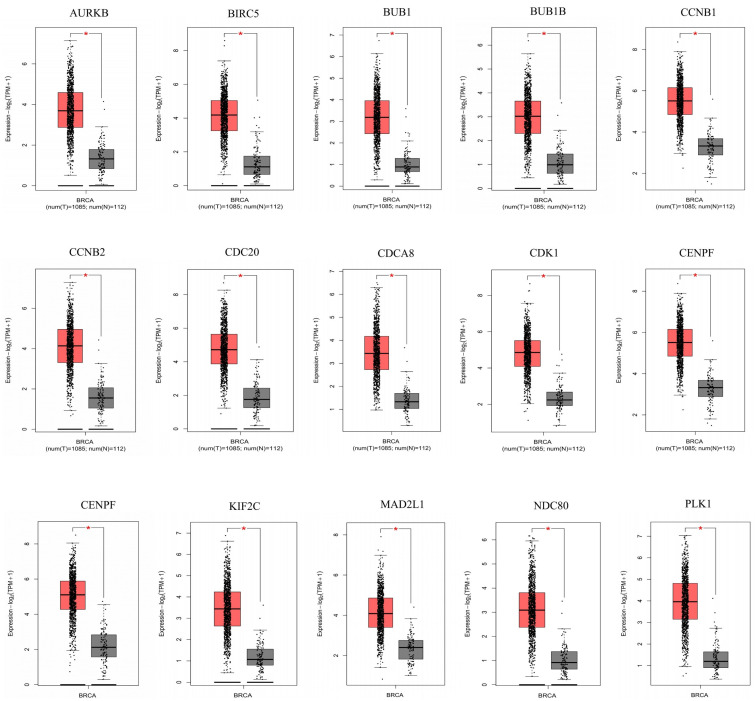
PPI The expression level of 15 hub genes in breast cancer tissue samples and normal tissue samples obtained from GEPIA 2 database. The red column represents tumor tissues, the grey column represents normal tissue. A total of 1085 breast cancer tissue samples and 112 normal tissue samples were included to investigate each gene expression level. The y-axis is the logarithm of gene expression in the sample. * represents that the difference between normal tissue samples and cancer tissue samples, which is statistically significant (*p* < 0.05).

**Figure 5 genes-13-00654-f005:**
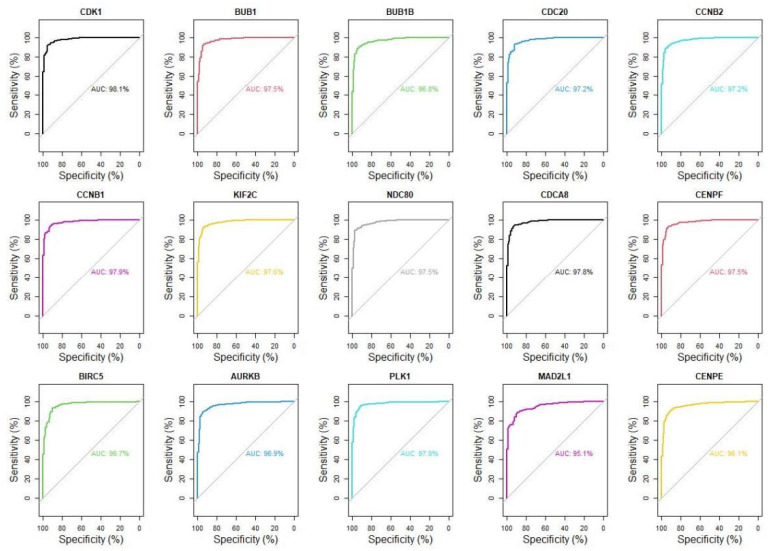
ROC curve of hub genes. The y-axis of ROC curve is sensitivity, the x-axis is specificity (equal to 1-false positive rate), and AUC value is the area under the curve.

**Figure 6 genes-13-00654-f006:**
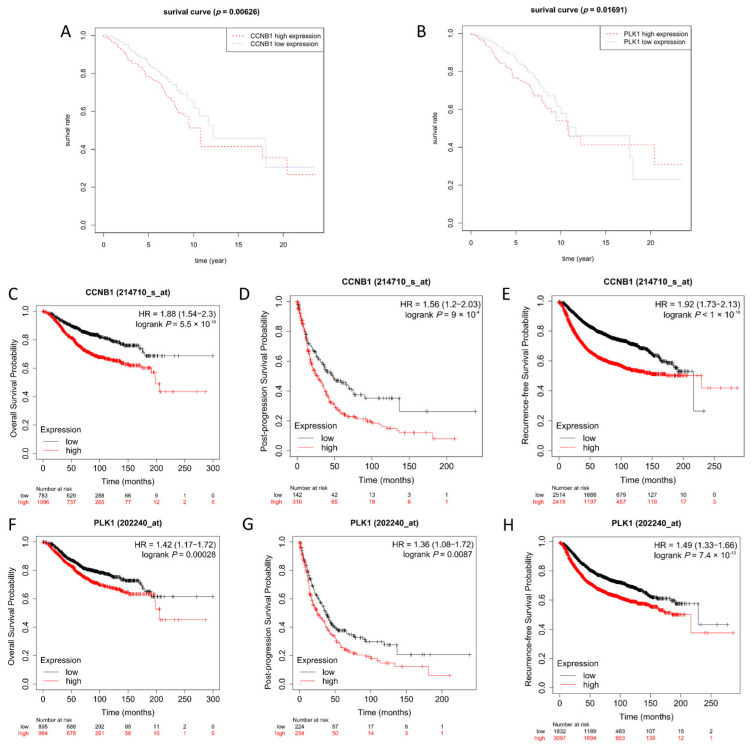
Kaplan−Meier survival analysis of the hub genes. (**A**) K−M analysis of *CCNB1*. (**B**) K–M analysis of *PLK1*. The x−axis is the survival time, and the death time or the last follow−up time of the patient was taken as the final survival time of the patients. The y−axis is survival rate. The samples were divided into “high” (Red) and “low” (Blue) gene expression groups, and the segmentation threshold was the median. *p* < 0.05 was considered statistically significant. (**C**) Overall survival rate (OS) of *CCNB1*. (**D**) Post-progression survival (PPS) rate of *CCNB1*. (**E**) Recurrence-free survival (RFS) rate of *CCNB1*. (**F**) Overall survival (OS) rate of *PLK1*. (**G**) Post-progression survival (PPS) rate of *PLK1*. (**H**) Recurrence-free survival (RFS) rate of *PLK1*. The upper x−axis is the survival time, and the y−axis is the survival rate. The samples were divided into “high” (Red) and “low” (Black) gene expression groups, and the segmentation threshold was auto select best cutoff. *p* < 0.05 was considered statistically significant.

**Figure 7 genes-13-00654-f007:**
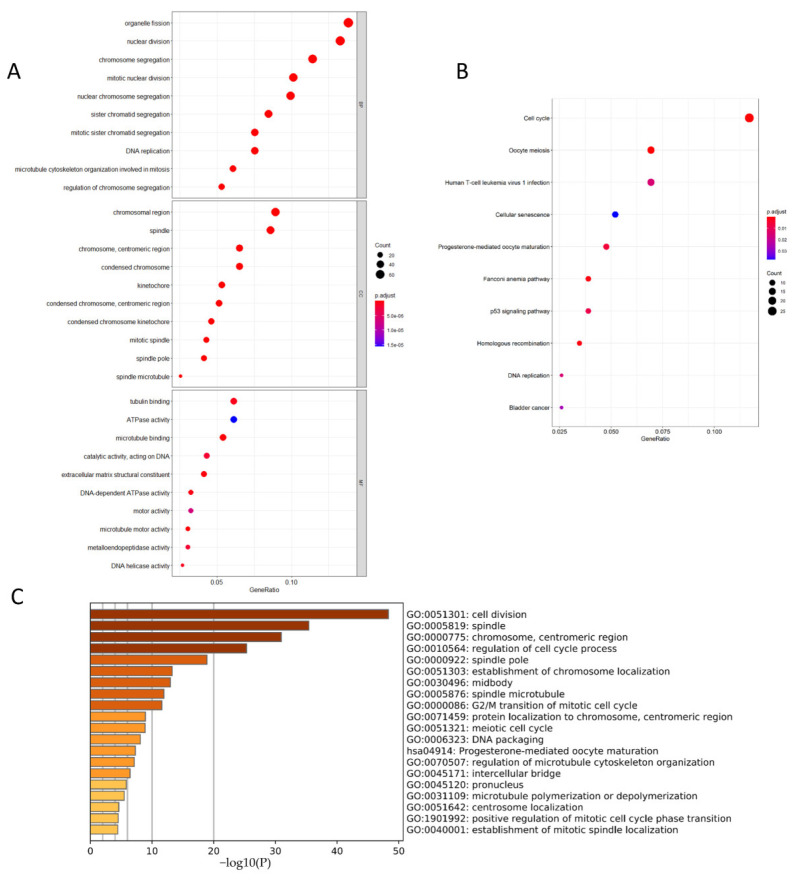
Enrichment analysis. (**A**) GO enrichment analysis of 693 module genes obtained from WGCNA. The enrichment results are carried out from three aspects: biological process (BP), cellular component (CC), and molecular function (MF). (**B**) The 693 module genes obtained from WGCNA were analyzed by KEGG enrichment analysis. The x−axis of each part of a−b is the proportion of genes enriched in this pathway of the total analyzed genes. The different size of the dot represents the number of genes enriched in the pathway, and the color of the dot represents the *p* value. *p* < 0.05 is regarded as statistically significant. (**C**) Enrichment analysis of 150 genes expression, which were similar to hub genes expression pattern. Each column bar on the y−axis represents an enrichment pathway, and the x-axis is the significance.

## Data Availability

The data that support the findings of this study are available in The Cancer Genome Atlas (TCGA) at [https://portal.gdc.cancer.gov/, accessed on 6 March 2022]. These data were derived from the following resources available in the public domain: [https://portal.gdc.cancer.gov/repository?facetTab=files&filters=%7B%22op%22%3A%22and%22%2C%22content%22%3A%5B%7B%22op%22%3A%22in%22%2C%22content%22%3A%7B%22field%22%3A%22cases.primary_site%22%2C%22value%22%3A%5B%22breast%22%5D%7D%7D%2C%7B%22op%22%3A%22in%22%2C%22content%22%3A%7B%22field%22%3A%22cases.project.program.name%22%2C%22value%22%3A%5B%22TCGA%22%5D%7D%7D%2C%7B%22op%22%3A%22in%22%2C%22content%22%3A%7B%22field%22%3A%22cases.project.project_id%22%2C%22value%22%3A%5B%22TCGA-BRCA%22%5D%7D%7D%2C%7B%22op%22%3A%22in%22%2C%22content%22%3A%7B%22field%22%3A%22files.data_category%22%2C%22value%22%3A%5B%22transcriptome%20profiling%22%5D%7D%7D%2C%7B%22op%22%3A%22in%22%2C%22content%22%3A%7B%22field%22%3A%22files.data_type%22%2C%22value%22%3A%5B%22Gene%20Expression%20Quantification%22%5D%7D%7D%2C%7B%22op%22%3A%22in%22%2C%22content%22%3A%7B%22field%22%3A%22files.experimental_strategy%22%2C%22value%22%3A%5B%22RNA-Seq%22%5D%7D%7D%5D%7D, accessed on 6 March 2022].

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
