# Peer review of "Bioinformatics Analysis Highlight Differentially Expressed CCNB1 and PLK1 Genes as Potential Anti-Breast Cancer Drug Targets and Prognostic Markers"

_genes, 2022, doi:10.3390/genes13040654_

Round 1
Reviewer 1 Report
The manuscript is based on published database to obtain the data and re-analyzed the data. After screening the data with several software, the authors suggest that CCNB1 and PLK1 genes are the potential anti-breast cancer drug targets.
There are many concerns with this manuscript.
1, it is difficult to understand many sentences , the authors need to improve the English significantly.
2, In the "Materials and Methods" part, the numerical labeling of 2.1 and 2.10 needs to be changed. they cannot be labeled as 1-3, then 4-6.
3, Figure 2A-C had better moving to supplementary Figures. reader cannot really get any information.
4, Figure 5 had better move to Supplementary Figures. This figure does not provide any useful information.
5, If the authors can find qRT-PCR data to confirm the expression of CCNB1 and PLK1 in cells, it will be better than immunohistochemical staining data.
6, In the discussion part, the authors need to add the content about the biological function of CCNB1 and PLK1 gene and the reports of these genes related to cancer, especially breast cancer.
In this manuscript, the authors based on available database to reanalyze the data to identify the predictive diagnosis and prognosis biomarkers and drug targets for breast cancer prevention. They identified that CCNB1 and PLK1 genes are highly expressed in all breast cancer stages and indicated that CCNB1 and PLK1 are potential drug targets and prognostic markers.
Some concerns:
1, The English writing needs to be improved. There are many grammatical errors.
2, “Materials and Methods” part: in 2.1: numerical labeling 1-3 is ok. However, in 2.10 numerical labeling 4-6. Don’t know what is the reason. It should be also labeled as 1-3.For “expression analysis”, please provide detail information to show which tools were used to quantify the protein expression.
3, Figure 2A-C, readers cannot really get any information from these figures, it had better move them into supplementary figures.
4, Figure 5 did not provide some useful information, it had better move to supplementary figures.
5, The legend of figure 5 and 6 may be mixed. the legend of fig 5 should be fig 6 and fig 6 should be fig 5.
Reviewer 2 Report
The manuscript by Fang and colleagues investigated 15 cellular factors that were reported to be correlated with breast cancer. In this paper, the authors performed gene expression analysis and correlated the expression data with clinical phenotypes in breast cancer. Then the authors focused on CCNB1 and PLK1 which are highly correlated with breast cancer and the survival analysis base on the gene expression was performed.
Line 74: Please state the normalization method for gene expression measurements, raw read count, RPKM, TPM or any other methods.
Line 77 and Line 101: The authors used both GDC/TCGA dataset and gene expression data from GEPIA2 database for DEG analysis, the difference between these two methods should be clarified.
Line 123: GO should be spelled out
Line 148: “The similarity of the first principal component was 19.9%,” it is better to say “first and principal components explained 19.9% of the total variance”
Line 152: “However, 152 these three methods have their shortcomings.” Reference for it
Line 160: Figure 1 legend “Schemes follow the same formatting.” What does it refer to?
Figure 1A, that “Dim1” “Dim2” refer to should be stated; Figure1C, x-axis, esgeR should be edgeR/
Line 162: Remove duplicate lines “The red circle represents the normal tissue, and the blue triangle 163 represents the breast cancer tissue”.
Line 164: “The abscissa is the similarity between the first principal component of cancer tissue sample and normal tissue sample, and the ordinate is the similarity between 165 the second principal component” abscissa or x-axis? Ordinate or y-axis?
Line 166: “The indistinguishable distinction between normal tissues and cancer tissues is due to the existence of adjacent tissues.” Reference for it
Line 178: The authors stated that 1209 samples were used for WGCNA analysis while only 1102 tumor samples were enrolled (line 68), dose normal samples also included in WGCNA analysis? If so, it may therefore produce a skewed result in line 202 and the correlation between the gene expression and breast cancer infection status would always be observed since only differentially expressed genes (DEGs) that obtained from differential expression analysis were used for correlation analysis.
Line 182 “weight (10-990),” what 10-990 refer to
Line 194 “removed in subsequent analysis.” Or “removed from subsequent analysis.”
Line 205: Figure 2E x-axis M, N, T, that they refer to should be stated
Line 208: Figure 2B y-axis R^2 should be consistent with the figure legends Line 208, Line 212.
Line 259 “However, no significant difference was observed 259 for the expression of CDC20 and KIF2C genes.”. Does that mean other genes showed the significant difference between tumor and normal samples? If so, please state the statistical method and show the P value between tumor and normal samples.
Line 267: Figure 5 and Figure 6 should be swapped
Line 288: “However, the other 13 genes were not statistically significant (P < 0.05)” P<0.05 or P>0.05
Line 298: “CCNB1 high-expression group”, “CCNB1 low-expression group”, “PLK1 high-expression group” and “PLK1 low-expression group”, how these groups were defined should be clarified.
Line 302 “which further suggested the importance of both genes as prognostic mark-302 ers and therapeutic targets.” The authors should state how CCNB1 or PLK1 could be the therapeutic targets since result data support this statement
Line 307 “were higher than the patients in other stages” should be “were higher than that in patients in other stages”
Line 364 “Enrichment analysis of 150 genes expression,” how these 150 genes were chosen should be clarified.
Line 404 “This may be due 404 to the sample size of CDC20 and KIF2C in the HPA database being too small.” Does it correct? Could the inconsistencies be caused by the post-transcriptional regulation or any other mechanisms?
Round 2
Reviewer 1 Report
The authors solved most of my concerns. I agree to accept it.